# Temperature dependence of quasiparticle interference in $d$-wave superconductors

Harun Al Rashid[1], Garima Goyal[1], Alireza Akbari[2] and Dheeraj Kumar Singh[1*]

**1** School of Physics and Materials Science,
Thapar Institute of Engineering and Technology, Patiala 147004, Punjab, India
**2** Max Planck Institute for the Chemical Physics of Solids, D-01187 Dresden, Germany

⋆ dheeraj.kumar@thapar.edu

## Abstract

We investigate the temperature dependence of quasiparticle interference in the high $T_c$-cuprates using an exact-diagonalization + Monte-Carlo based scheme to simulate the $d$-wave superconducting order parameter. The quasiparticle interference patterns have features largely resulting from the scattering vectors of the octet model at lower temperature. Our findings suggest that the features of quasiparticle interference in the pseudogap region of the phase diagram are also dominated by the set of scattering vectors belonging to the octet model because of the persisting antinodal gap beyond the superconducting transition $T_c$. However, beyond a temperature when the antinodal gap becomes very small, a set of scattering vectors different from those belonging to the octet model are responsible for the quasiparticle interference patterns. With a rise in temperature, the patterns are increasingly broadened.

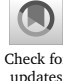

# 1 Introduction

The appearance of a *pseudogap* phase at higher temperature remains a puzzling feature of the phase diagram of high-$T_c$ cuprates [1–6]. A significant part of our understanding of this phase is based on the scanning-tunneling microscopy (STM) and angle-resolved photoemission spectroscopy (ARPES) [7–18]. Quasiparticle interference (QPI) determined by the STM has become one of the most powerful techniques in recent times for exploring the electronic properties of different phases of various correlated electron systems [19, 20]. Not only can the QPI reveals the electronic structure in the vicinity of the Fermi surface [21–25] but it may also be used as a phase-sensitive tool to investigate the nature of superconducting order parameter [26–35].

The structure of QPI patterns is determined by three major factors. First of all, the basic features of the patterns result from the topology of the constant energy contours (CEC) [24, 26, 36]. Secondly, they are also dependent on the nature of scatterers. For instance, a magnetic or a non-magnetic impurity potential may give rise to entirely different patterns especially when the order parameter changes sign in the Brillouin zone [30]. For a magnetic impurity in the superconducting state, the pattern is dominated by those scattering vectors which connect the parts of CECs with the same sign of the superconducting order parameter. While in the case of non-magnetic impurities, the QPI patterns have enhanced features corresponding to the scattering vectors connecting the sections of CECs having opposite signs of the order parameter [26]. Furthermore, the antisymmetrized local density of states (LDOS) summed over momenta can show very different frequency-dependent behavior for a sign changing and preserving superconducting order parameter [34]. Thirdly, there will also be a larger modulation in the DOS for the scattering vectors joining the sections of CECs with a high spectral density [24].

The CECs in the $d$-wave superconducting state assume the shape of a bent ellipse which is similar to the cross-section of a banana when cut along its length [23]. The predominant features of the patterns in a $d$-wave superconductor are widely believed to be captured within the so-called *octet model*. There are altogether seven scattering vectors which connect the tips of the bananas' cross-section and are expected to play a crucial role in the formation of QPI patterns [26, 37, 38]. However, it is not clear as to what will be the pseudogap-specific fingerprints in the QPI patterns across the superconducting transition. Only a few experimental studies have been carried out because of the difficulties associated with the rapid growth of thermal fluctuation in the energy of a tunneling electron with increasing temperature [39] and the scenario remains largely similar with regard to theoretical studies.

The pseudogap phase is marked by a dip in the density of states persistent above $T_c$ up to a temperature $T^*$ [9]. The shape of the dip is similar to the $d$-wave gap, but it is comparatively shallower, [11] and accompanied by the Fermi arc formation. The Fermi arc size increases with temperature [4, 15] and the normal state Fermi surface is recovered at the onset temperature $T^*$ for the pseudogap phase, whereas the Fermi points are protected against thermal phase fluctuations below $T_c$ [14]. The CECs for the finite non-zero energy may exhibit characteristics within the temperature range $T_c < T < T^*$ similar to those below $T_c$ and therefore the QPI patterns are expected to carry largely similar features [39].

In this paper, we examine the QPI features of the pseudogap phase at finite temperature, we adopt an approach based on the exact diagonalization + Monte Carlo (ED+MC) scheme, which was used previously to capture several spectral characteristics of the pseudogap phase within a minimal model of $d$-wave superconductors [40]. The momentum-resolved spectral function for a larger system was possible to study with the approach. Two-peak structures with a shallow dip for zero energy on approaching the antinodal point were found to exist beyond the superconducting transition temperature indicating the loss of long-range phase

correlation. Thermally equilibrated superconducting order parameter on the lattice points are used to construct temperature-dependent Green's function, which, then, can be used to calculate Green's function modulated by the presence of an impurity potential.

The structure of the paper is as follows. In section II, we discuss the model and provide the details of methodologies. Section III is used to discuss the major results whereas we conclude in section IV.

## 2  Model and Method

We consider the following one-band effective Hamiltonian

$$H_{eff} = -\sum_{\mathbf{i},\delta',\sigma} t_{\mathbf{i},\mathbf{i}+\delta'} d_{\mathbf{i}\sigma}^{\dagger} d_{\mathbf{i}+\delta'\sigma} - \mu \sum_{\mathbf{i}} n_{\mathbf{i}} - \sum_{\mathbf{i},\delta} \left( d_{\mathbf{i}\uparrow}^{\dagger} d_{\mathbf{i}+\delta\downarrow}^{\dagger} + d_{\mathbf{i}+\delta\uparrow}^{\dagger} d_{\mathbf{i}\downarrow}^{\dagger} \right) \Delta_{\mathbf{i}}^{\delta} + H.c. + \frac{1}{V} \sum_{\mathbf{i}} \left| \Delta_{\mathbf{i}}^{\delta} \right|^{2} . \quad (1)$$

The first term describes the kinetic energy term originating from the first and second nearest neighbor hoppings. $t$ and $t'$ are the corresponding hopping parameters. $d_{\mathbf{i}\sigma}^{\dagger}(d_{\mathbf{i}\sigma})$ creates (destroys) an electron at site $\mathbf{i}$ with spin $\sigma$. $\delta'$ has been used for the position of both the first and second nearest neighbors whereas $\delta$ denotes the first neighbor only. The unit of energy is set to be $t$ throughout and $t' = $ -0.4 [12]. In the second term, $\mu$ is the chemical potential, which has been chosen to correspond to $n \sim 0.9$. The third term is the bilinear term in the electron field operators and the fourth term is a scalar term. The temporal fluctuations of the order parameters are ignored and the spatial fluctuations are retained. The fourth term, ignored in the Hartree-Fock meanfield theoretic approach, plays a critical role in the simulation carried out here. The last two terms of the Hamiltonian are obtained by using the Hubbard-Stratanovich transformation in the $d$-wave pairing channel of the nearest-neighbor attractive interaction, while ignoring other channels such as charge-density etc.

We focus on the interaction parameter $V \sim 4t^2/U \approx 1$ though a larger value will further increase the temperature window of pseudogap phase as noted earlier. We set $V \sim 1.2$. The $d$-wave order parameter $\Delta_{\mathbf{i}}^{\delta} = \langle d_{\mathbf{i}\downarrow} d_{\mathbf{i}+\delta\uparrow} \rangle = |\Delta_{\mathbf{i}}| e^{i\phi^{\delta}}$ is treated as a complex classical field. It is a link variable and can be expressed as a product of two variables, one of them is a site variable $|\Delta_{\mathbf{i}}|$ and other is a link variable $\phi^{\delta}$ ($\delta = x, y$). The equilibrium configuration $\{\Delta_{i}, \phi_{i}^{x}, \phi_{i}^{y}\}$ of the amplitude and phase fields are obtained by the Metropolis algorithm in accordance with the distribution $\{\Delta_{i}, \phi_{i}^{x}, \phi_{i}^{y}\} \propto \mathrm{Tr}_{dd^{\dagger}} e^{-\beta H_{eff}}$.

Here, we use a combination of two techniques, $i.$ $e.$, "traveling-cluster approximation (TCA) [41] + twisted-boundary condition (TBC)" [42] to examine the changes in the spectral function resulting from the impurities in a system of large size. First, the equilibrated configuration for $14 \times 14$ lattice size is obtained with the help of a traveling cluster of size $6 \times 6$. Secondly, TBCs are used along the $x$ and $y$ directions in such a way that it becomes equivalent to repeating an equilibrated configuration $N_s = 12$ times along both directions. In other words, the impurity induced modulation in the spectral function is obtained by using Bloch's theorem for an effective lattice of size $168 \times 168$.

In the earlier work [40], the temperature-dependent behavior of the single-particle spectral function within the minimal model of $d$-wave given by Eq. 1 was examined. For $V \sim 1$, as also considered in the current work, the long-range phase correlation was found to develop near $T \sim T_c$, whereas the short-range phase correlation continued to exist up even up to higher temperature. The antinodal gap in the form of two-peak structure with a shallow dip near $\omega = 0$ was shown to exist beyond the superconducting-transition temperature $T_c$. For $T < T_c$, there exists a Fermi surface in the form of Fermi points against thermal phase fluctuations whereas all the non-nodal points in the normal-state Fermi surface are accompanied with a two-peak spectral features with a dip at $\omega = 0$. Beyond $T_c$, the Fermi points are transformed

into arcs, characterized by a single quasiparticle peak. With an increasing temperature, the Fermi arcs grow in size resulting into the recovery of the normal state Fermi surface at a temperature $T^* > T_c$. For $V \sim 1$, it was found that $T^* \sim 1.5 T_c$.

In order to calculate the modification introduced into the Green's function by a single-impurity atom, we require the bare Green's function. It can be obtained by using the real-space complex classical fields configuration corresponding to the $d$-wave superconductivity as below

$$\hat{G}^0(\mathbf{k},\omega) = \sum_{\alpha l} \frac{1}{\omega - E_{\alpha l} + i\eta} \begin{pmatrix} |\langle u_{\alpha l}|\mathbf{k}\rangle|^2 & -\langle u_{\alpha l}|\mathbf{k}\rangle^* \langle v_{\alpha l}|\mathbf{k}\rangle^* \\ -\langle u_{\alpha l}|\mathbf{k}\rangle \langle v_{\alpha l}|\mathbf{k}\rangle & |\langle v_{\alpha l}|\mathbf{k}\rangle|^2 \end{pmatrix}$$
$$+ \sum_{\alpha l} \frac{1}{\omega + E_{\alpha l} + i\eta} \begin{pmatrix} |\langle v_{\alpha l}|\mathbf{k}\rangle|^2 & \langle u_{\alpha l}|\mathbf{k}\rangle^* \langle v_{\alpha l}|\mathbf{k}\rangle^* \\ \langle u_{\alpha l}|\mathbf{k}\rangle \langle v_{\alpha l}|\mathbf{k}\rangle & |\langle u_{\alpha l}|\mathbf{k}\rangle|^2 \end{pmatrix}, \qquad (2)$$

where $|u_\alpha l\rangle$ and $|v_\alpha l\rangle$ form the eigenvectors of the Bogoliubov−de Gennes Hamiltonian corresponding to the eigenvalues $E_{\alpha l}$. The subscript $\alpha$ indicates a particular lattice site while $l$ identifies a particular lattice in the superlattice structure. Thus, the spectral function is obtained as

$$A(\mathbf{k},\omega) = \sum_{\mathbf{q},\lambda} \left( \left| \langle \mathbf{k}|u_{\mathbf{q},\lambda}\rangle \right|^2 \delta(\omega - E_{\mathbf{q},\lambda}) + \left| \langle \mathbf{k}|v_{\mathbf{q},\lambda}\rangle \right|^2 \delta(\omega + E_{\mathbf{q},\lambda}) \right),$$

where

$$\langle \mathbf{k}|u_{\mathbf{q},\alpha}\rangle = \sum_l \sum_i \langle \mathbf{k}|l,i\rangle \langle l,i|u_{\mathbf{q},\lambda}\rangle, \qquad (3)$$

$l$ is the superlattice index and $i$ is a site index within the superlattice.

Impurity-induced contribution to the Green's function based on the perturbation theory is given by [30, 31, 36]

$$\delta \hat{G}(\mathbf{k},\mathbf{k}',\omega) = \hat{G}^0(\mathbf{k},\omega) \hat{T}(\omega) \hat{G}^0(\mathbf{k}',\omega). \qquad (4)$$

$\hat{G}^0(\mathbf{k},\omega) = (\hat{\mathbf{I}} - \hat{H}(\mathbf{k}))^{-1}$ is the bare Green's function. $\hat{H}(\mathbf{k})$ is the Hamiltonian in the absence of any impurity and $\hat{\mathbf{I}}$ is a 2×2 identity matrix. The matrix $\hat{T}(\omega)$ is given by

$$\hat{T}(\omega) = (\hat{\mathbf{I}} - \hat{V}_{n/m} \hat{\mathcal{G}}(\omega))^{-1} \hat{V}_{n/m}, \qquad (5)$$

where

$$\hat{\mathcal{G}}(\omega) = \frac{1}{N} \sum_{\mathbf{k}} \hat{G}^0(\mathbf{k},\omega). \qquad (6)$$

The matrix $\hat{V}_{n/m}$ is

$$\hat{V}_{n/m} = V_\circ \begin{pmatrix} 1 & 0 \\ 0 & \mp 1 \end{pmatrix}, \qquad (7)$$

$V_n$ and $V_m$ are the matrices for nonmagnetic and magnetic impurities. We set the impurity scattering strength $V_\circ \sim 0.1$. $g$-map or the fluctuation $\delta N(\mathbf{q},\omega)$ in the LDOS due to a delta-like impurity scatterer is given by

$$\delta N(\mathbf{q},\omega) = \frac{i}{2\pi} \sum_{\mathbf{k}} g(\mathbf{k},\mathbf{q},\omega), \qquad (8)$$

with

$$g(\mathbf{k},\mathbf{q},\omega) = \sum_i \left( \delta \hat{G}^{ii}(\mathbf{k},\mathbf{k}',\omega) - \delta \hat{G}^{ii*}(\mathbf{k}',\mathbf{k},\omega) \right), \qquad (9)$$

where $\mathbf{k} - \mathbf{k}' = \mathbf{q}$.

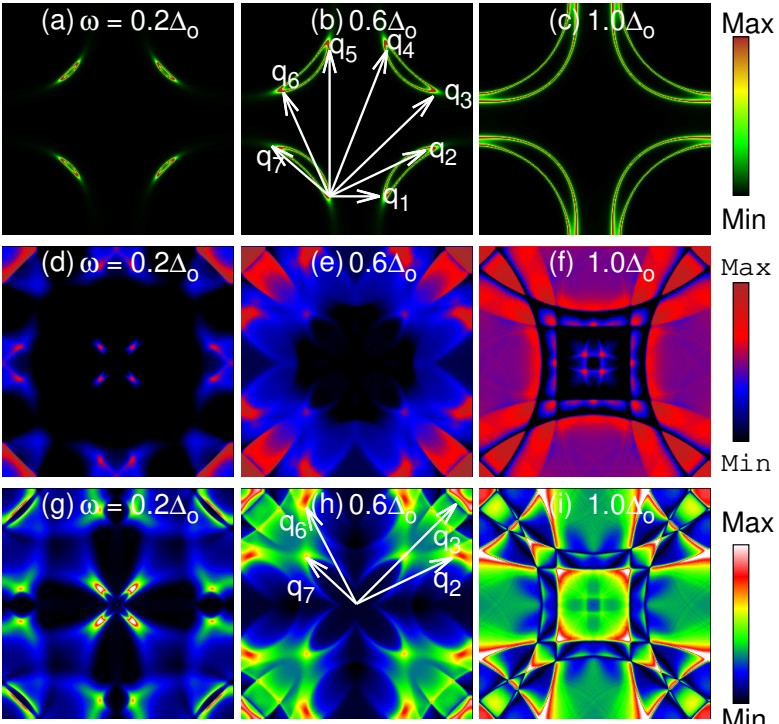

Figure 1: (a-c) The CECs for the quasiparticle energies in steps of $0.4\Delta_o$ within the range $0 \leq \omega \leq \Delta_o$. The banana-shaped CECs can be seen to exist up to $\Delta_o$. (d-f) $g$-maps and (g-i) $z$-maps for various quasiparticle energies in similar steps when a non-magnetic impurity is present. The QPI patterns are dominated by the scattering vectors $\mathbf{q}_2$, $\mathbf{q}_3$, $\mathbf{q}_6$, and $\mathbf{q}_7$. There is no significant difference between the two sets of map.

The tunneling-matrix element is dependent on the distance between the tip and sample. However, $Z(\mathbf{q}, \omega)$, a quantity independent of tunneling-matrix element between the STM tip and sample surface, is defined as the following ratio

$$Z(\mathbf{r}, E) = \frac{N(\mathbf{r}, E)}{N(\mathbf{r}, -E)}. \tag{10}$$

The Fourier transform of $Z(\mathbf{r}, E)$ is obtained as

$$Z(\mathbf{q}, E) \approx \frac{N^0(\omega)}{N^0(-\omega)} \left[ \frac{\delta N(\mathbf{q}, \omega)}{N^0(\omega)} - \frac{\delta N(\mathbf{q}, -\omega)}{N^0(-\omega)} \right], \tag{11}$$

where $\delta(\mathbf{q})$ is not included and $N^0(\omega)$ is the density of states. Various quantities mentioned above are obtained by calculating their average over different field configurations at a given temperature.

The QPIs in the $d$-wave state can also be examined at low temperature by ignoring the scalar term in the Hamiltonian given by Eq. 1 and by Fourier transforming the electron creation and annihilation operator, which leads to

$$\mathcal{H} = \sum_{\mathbf{k}} \Psi^\dagger(\mathbf{k}) \hat{H}(\mathbf{k}) \Psi(\mathbf{k})$$

$$= \sum_{\mathbf{k}} \Psi^\dagger(\mathbf{k}) \begin{pmatrix} \varepsilon(\mathbf{k}) & \Delta_{\mathbf{k}}^\dagger \\ \Delta_{\mathbf{k}} & -\varepsilon(\mathbf{k}) \end{pmatrix} \Psi(\mathbf{k}), \tag{12}$$

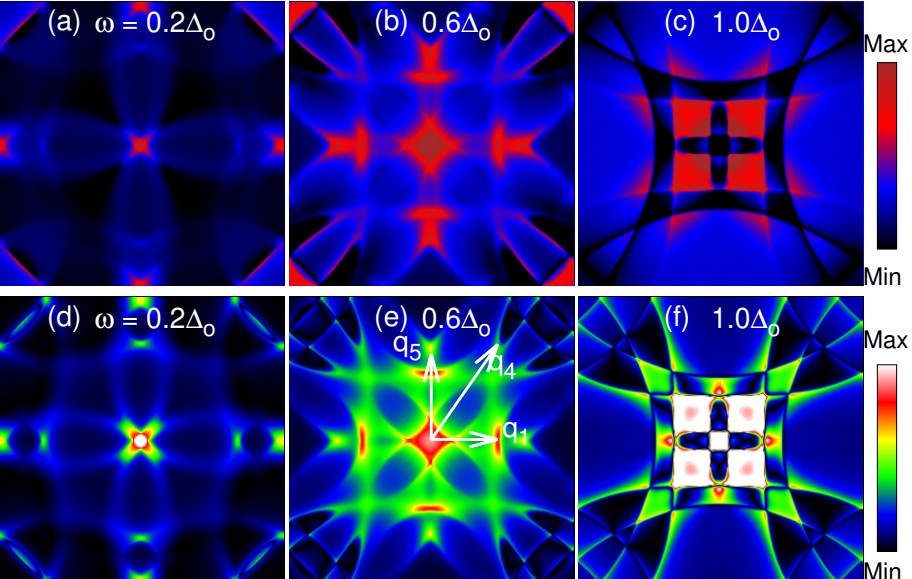

Figure 2: (a-c) $g$-maps and (d-f) $z$-maps for the quasiparticle energies in steps of $0.4\Delta_o$ when the impurity atom is magnetic. The patterns have enhanced features corresponding to the scattering vectors $\mathbf{q}_1$, $\mathbf{q}_4$, and $\mathbf{q}_5$.

where the electron-field operator is defined within the Nambu formalism as $\Psi^\dagger_{\mathbf{k}\uparrow} = (d^\dagger_{\mathbf{k}}, d^\dagger_{-\mathbf{k}})$.
$\varepsilon(\mathbf{k}) = -2t(\cos k_x + \cos k_y) + 4t' \cos k_x \cos k_y$ . $\Delta(\mathbf{k}) = \Delta_o(\cos k_x - \cos k_y)/2$.

## 3 Results and Discussion

Figs. 1(a-c) show the CECs in the $d$-wave superconducting state as a function of quasiparticle energy. The banana-shaped CECs exist when $\omega \lesssim \Delta_0$, where $\Delta_0 \sim 0.5$. We find that the spectral density along the CECs are peaked at the tips. As a result, the scattering vectors joining the tips are expected to dominate the QPI patterns. There are altogether eight such tips in the set of four Fermi surfaces and joining of these tips results into seven scattering vectors giving rise to the so-called *octet* model (See Fig. 4 also).

Figs. 1(d-f) show the $g$-map for the quasiparticle energy in steps of $\Delta\omega = 0.4\Delta_0$ when the impurity scatterers are non-magnetic in nature. The QPI patterns for $\omega \lesssim 0.5\Delta_0$ are dominated by the scattering vectors $\mathbf{q}_2, \mathbf{q}_3, \mathbf{q}_6$, and $\mathbf{q}_7$. This is mainly because they join those parts of CECs where the superconducting order parameter has the opposite signs. The coherence factor is expected to get significantly suppressed for those $\mathbf{q}_i$s which connect the regions with the order parameters having the same sign. The features due to $\mathbf{q}_2, \mathbf{q}_3, \mathbf{q}_6$, and $\mathbf{q}_7$ can be seen clearly when $\Delta\omega \sim 0.5\Delta_0$. On increasing $\omega$ further until $\omega \lesssim \Delta_0$, these features approach each other. Figs. 1(g-i) show the corresponding $z$-map. We find the $z$-map to be qualitatively similar to the $g$-map except only a few minor differences until $\omega \lesssim \Delta_0$. The differences are significant only beyond $\omega \sim \Delta_0$ (not shown here).

Figs. 2 (a-c) and (d-f) show the QPI patterns when the impurity atoms are magnetic. For small $\omega$, the patterns due to both $\mathbf{q}_1$ and $\mathbf{q}_5$ almost coincide near $(\pi, 0)$. With an increase in $\omega$, the patterns shift towards $(0, 0)$ along the line joining the points $(0,0)$ and $(\pi, 0)$. The features due to $\mathbf{q}_5$ are found near $(\pi, \pi)$, which continue to shift towards the line joining the points $(0, 0)$ and $(\pi, 0)$, and merge finally on approaching $\omega \sim \Delta_0$. Note the suppression of

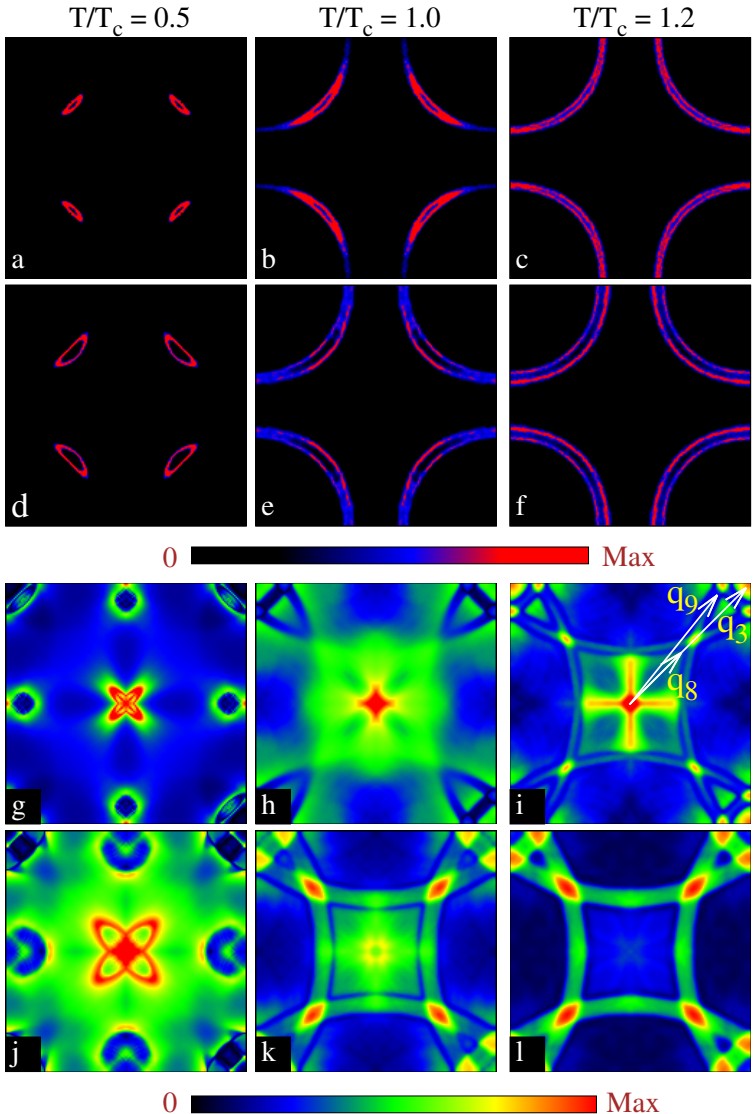

Figure 3: Evolution of quasiparticle spectral function $\mathcal{A}(\mathbf{k}, \omega)$ for $\omega =$ (a-c) $0.15t$ and (d-f) $0.3t$ as a function of temperature. $k_x$ and $k_y$ are along horizontal and vertical directions, respectively, with each having range $[-\pi, \pi]$. Corresponding QPIs for $\omega =$ (g-i) $0.15t$ and (j-l) $0.3t$.

patterns due to scattering vectors such as $\mathbf{q}_2$, $\mathbf{q}_7$ etc. as they don't connect sections of CECs having sign of $d$-wave superconducting order parameters opposite to each other.

The major features of the spectral properties of single-particle spectral functions, using the approach described in the current work, have been shown earlier to be in agreement with ARPES measurements [15, 40] . The Fermi points exist up to the superconducting transition temperature $T_c$ although the phase fluctuations do result in the spectral weight transfer from the nodal to nearby points. Beyond $T_c$, the Fermi arcs appear, and they exist up to $T^*$ the onset temperature of the pseudogap phase. We examine $\mathcal{A}(\mathbf{k}, -\omega) + \mathcal{A}(\mathbf{k}, \omega)$ first. Figs. 3(a)-(f) show the CECs for the quasiparticle energy $\omega = 0.15$ and $0.30$. The antinodal gap $\Delta_{an}(T)$ survives up to $T_c$ and beyond. $\Delta_{an}(0) \sim 2\Delta_{an}(T_c)$ while $\Delta_{an}(T_c) \sim 0.5t$. The banana-shaped CECs can be seen up to $\omega \sim \Delta_{an}(T_c)$ for $T = 0.5T_c$ while the same is not true for $T \sim T_c$. With the gap near the nodal points about getting filled, the banana-shaped CECs

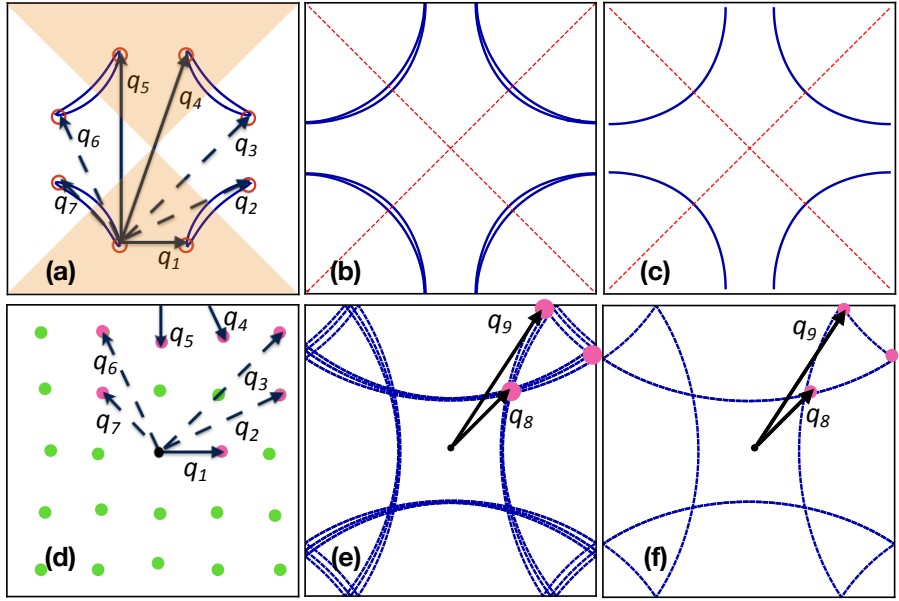

Figure 4: (a) The schematics of the banana-shaped CECs in the $d$-wave supercon-ducting state. The CECs at non-zero (b) finite energy $\omega \gtrsim \Delta_{an}$ in the temperature regime $T_c < T < T^*$ or when $T > T^*$ and (c) $\omega = 0$. The scattering vectors $\mathbf{q}_8$ and $\mathbf{q}_9$ are mainly responsible for the QPI patterns at high energy and high temperature. (d), (e) and (f) indicate scattering vectors and corresponding patterns.

are highly elongated and rather look like the FSs in the normal state.

Figs. 3(g)-(l) show the QPI patterns $Z(\mathbf{q}, \omega)$ in the presence of non-magnetic impurity. At low energy and temperature, the patterns are clearly dominated by the three set of scattering vectors $\mathbf{q}_2$, $\mathbf{q}_3$, and $\mathbf{q}_7$ (from the "octet" model). It should be noted that $\mathbf{q}_6$ can be obtained by a $\pi/2$ rotation of $\mathbf{q}_2$. We first examine the QPI patterns for $T/T_c = 0.5$. When $\omega \sim 0.5\Delta_{an}(T_c)$, the dominant patterns in the QPI owing mainly to the scattering vectors $\mathbf{q}_2$ (or $\mathbf{q}_6$), $\mathbf{q}_3$, and $\mathbf{q}_7$ can easily be identified. As quasiparticle energy increases to $\omega = \Delta_{an}(T_c)$, the size of $\mathbf{q}_7$ increases as well, leading to an enlarged four-petaled flower-like pattern around $(0,0)$. Note that the average size of $\mathbf{q}_2$ remains almost the same, and only a weak semicircular pattern around $(0, \pm\pi)$ is seen. The signature of reduction in the $d$-wave order parameter with temperature is also reflected in the patterns for $T/T_c \sim 1$, which is anticipated because $\Delta_{an}(T_c) \sim 0.5\Delta_{an}(0)$. The patterns for $\omega \sim 0.5\Delta_{an}(T_c)$ near $T_c$ look similar to that for $\omega \sim \Delta_{an}(T_c)$ near $T/T_c \sim 0.5$, which is not surprising given the similar structure of the CECs.

In the pseudogap phase, *i.e.*, beyond $T_c$, an important difference from the normal state QPI pattern that is expected is the survival of the pattern corresponding to the scattering vectors such as $\mathbf{q}_3$ and $\mathbf{q}_7$. Fig. 3 (i) shows the corresponding patterns, which is purely originating from the persistence of the anti-nodal gap otherwise absent in the normal state. This is a consequence of the persisting antinodal gap persisting beyond $T_c$ up to $T^*$. In the vicinity of $T^*$ and beyond, the QPI patterns exhibit robust behavior against the change in temperature. These high-temperature features originate from the scattering vectors not belonging to the "octet" model. Instead, we find a set consisting of scattering vectors $\mathbf{q}_8$ and $\mathbf{q}_9$ that are responsible for the persistent features at elevated temperatures (Fig. 4). Note that the CECs have nearly a circular shape in the extended Brillouin zone, therefore, the dominant scattering vector will have magnitude twice of the radius of the circular Fermi surface. With this as a radius, if circles are drawn then the scattering vectors such as $\mathbf{q}_8$ and $\mathbf{q}_9$ with tips at the intersections of such

two circles will lead to the dominating features of the QPI. Clearly, the scattering vectors $\mathbf{q}_8$ and $\mathbf{q}_9$ are different from those of the "octet" model.

Recent works have highlighted the importance of using continuum Green's function within a Wannier basis in order to take into account the electronic cloud around the lattice point with an appropriate phase associated with a particular orbital such as $d_{x^2-y^2}$ orbital in cuprates [35, 43, 44]. QPI based on such a realistic electronic cloud around the lattice point provides a better description of the scanning-tunneling microscopy results. However, our focus, in this work, was to primarily examine the temperature dependence of QPI. The spatial distribution of the electronic cloud is going to be largely unaffected with a rise in temperature. Thus, the essential difference between the cases with and without the electronic clouds at low temperatures as shown in earlier work is expected to be present even at finite temperatures.

The pseudogap phase is a rather complex phase. Although it is widely believed that the pseudogap-like feature may originate from short-range magnetic correlations [45–47], signatures of short-range phase coherence [48, 49], nematic order without four-fold rotation symmetry revealed by the anisotropy in the magnetic susceptibility measurements [6], charge-density order [50], pair-density wave order [51] etc. have also been obtained. Therefore, a more realistic study requires additional terms in the Hamiltonian to account for such symmetry breaking and it will be interesting to see the associated features in the quasiparticle interference.

# 4   Conclusions

To conclude, we have examined the quasiparticle interference in a minimal model of high-$T_c$ cuprates. Our findings suggest that the low temperatures features of the quasiparticle interference agrees well with the octet model, which is reflected in both $g$- and $z$-map obtained due to magnetic or nonmagnetic impurities. There is no significant difference in $g$- and $z$-map, particularly when $\omega \leq \Delta_o$, whereas beyond $\omega = \Delta_o$, the differences may get enhanced. When the temperature increases, the size of the contour of constant energy surface increases because of a reduction in the antinodal gap. Accordingly, the patterns are modified . At further higher temperature, that is near the onset of pseudogap to $d$-wave superconducting transition temperature and beyond the quasiparticle interference pattern is dominated by two new scattering vectors in addition to the ones that connect antinodal regions.

# Acknowledgment

D.K.S. was supported through DST/NSM/R&D_HPC_Applications/2021/14 funded by DST-NSM and start-up research grant SRG/2020/002144 funded by DST-SERB. D.K.S. gratefully acknowledges Pinaki Majumdar for useful discussions. We are also grateful to Peter Thalmeier for fruitful comments and suggestions.

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
