# Peer review of "Temperature dependence of quasiparticle interference in $d$-wave superconductors"

_SciPost Physics Core, doi:SciPost Phys. Core 6, 033 (2023)_

## Round 1 · Referee Report · Anonymous (Referee 1) · 2022-11-9

Strengths

1) Good introduction and motivation of the topic.

2) Novel Hamiltonian.

Weaknesses

1) The authors do not present significant new insight into the question that they have posed.

Report

Rashid et al. report Quasiparticle Interference (QPI) simulations of a minimal model of a d-wave superconductor on a square lattice with nearest neighbour hopping. The novelty here appears in their new minimal model of d-wave superconductivity that additionally captures pseudogap physics, as seen in e.g cuprate systems.

The paper was nicely motivated and introduced, indeed it would be interesting in this minimal model of d-wave superconductivity, which already captures the pseudogap, to know how the experimental QPI signatures evolve as a function of temperature from such a model. However, I do not feel that the authors have sufficiently answered this question.

Most of the results and discussion are focused on the superconducting state (Fig. 1 and Fig. 2) Here the qualitative comparison between the octet model and QPI scattering is established, but this is not new information and has been discussed extensively within the literature since the early 2000’s [E.g. Mcelroy et al. (2003) -Ref 23 of the manuscript - or Pereg-Barnea et al. (2003)- Ref 37 of the manuscript]. Additionally, the “new” scattering vectors that the authors identify outside the superconducting state appear to be the expected QPI pattern from the normal state, which again is not new information [see e.g Kreisel et al. PRL 114, 217002 (2015) – not cited - or Gu et al. Nat. Comm., 10, 1603 (2019) – not cited].

The actual study of the consequence of the pseudogap on the QPI pattern appears to be missing. For example, How does the QPI pattern evolve at the Fermi level as a function of temperature? Are there clear signatures of the pseudogap in the QPI that aren't found in the normal state? Are there any changes to the real space QPI pattern as a function of temperature? From the two panels (Fig. 3(h) and 3(k)) it looks as if the pseudogap phase produces a similar QPI pattern to the normal state (Fig. 3f and 3(i)), but there is only limited information to tell if this is true or not. As this was the motivation of the study, I feel that the opportunity to discuss the new physics that this model contains was missed.

I also question why the authors did not use the continuum transformation of the Green's function to study the QPI pattern? Here the authors use a Green's function that in real space would sample the density of states once per unit cell. However STM measurements are continuous in real space. The Continuum QPI (cQPI) technique has become well established in recent years, and provides a much more accurate description of the QPI seen in STM (See for example Kreisel et al. (2015) - PRL 114, 217002 - or Boker et al. (2020) - Ref 35 of this manuscript - for cQPI calculations on models of superconducting cuprates.)

In summary, Whilst I appreciate the motivation of the study, and the novelty of the model that the authors employ, I am unsure what new physical results have been obtained that will be useful for future experiments or for the understanding of the pseudogap state. Therefore I do not believe this manuscript currently meets the criteria "Detail one or more new research results significantly advancing current knowledge and understanding of the field." required for publication in SciPost Physics Core.

However, if the authors can revise the manuscript - strengthen their discussion of the QPI patterns in the pseudogap state and focus their key findings - then I believe this could warrant a significant advancement of the field and be publishable.

Requested changes

1) The authors write that "the QPIs in the d-wave state can also be examined at low temperature by ignoring the scalar term in the Hamiltonian given in Eq. 1 and by Fourier transforming the electron creation and annihilation operator, which leads to (Eq 13)…” Here I am confused, are the calculations in the main text using the new Hamiltonian (Eq. 1) or Eq. 13. Because if they are using Eq. 13 then the QPI calculation has already been performed in multiple publications [see the four references cited above in the report for a non-exhaustive list]. If you did not use Eq. 13, then why did you write this in the methodology section? - Could you please clarify this point?

2) The authors refer to specific vectors (q1-q7) in the octet model throughout this text, but a definition of the vectors does not appear until Fig 4a. As a reader, I was confused as to what scattering vectors you were referring to in the QPI patterns. For example, in the caption of Fig.1 The authors write “The QPI patterns are dominated by the scattering vectors q2,q3,q6,q7” but it is not specified where q2,q3,q6,q7 belong on the g- or z-maps. Arrows highlighting the specific vectors would help the reader to understand what the authors mean.

3) In the results section (second paragraph page 3) “The features due to q2,q3,q6,q7 can be seen clearly when Delta w ~ 0.5 Delta. Do you mean when comparing energy layers spaced by 0.5 Delta? or do you mean when the energy is half the superconducting gap? Could you clarify what you mean?

4) In the final paragraph of page 3 “Note the suppression of patterns because of other scattering vectors for the very reason that they don’t connect sections of CEC having sign of d-wave superconducting order parameters opposite to each other”. Doesn’t make sense, and I am unclear what the authors intend to say here. - Could you clarify?

5) There is some repetition in the introduction, “In this paper, we examine the QPI features of pseudogap phase at finite temperature, … …. In this paper we study the temperature dependent behavior of QPIs in a minimal model of d-wave superconductivity”.

6) Figure 4, (d),(e) and (f) are not labelled in the caption nor are they referred to in the main text. - Could you please add these descriptions to the caption and discuss the panels in the text.

  • validity: good
  • significance: low
  • originality: low
  • clarity: ok
  • formatting: excellent
  • grammar: good

Author:  Dheeraj Kumar Singh  on 2022-12-19  [id 3155]

(in reply to Report 1 on 2022-11-09)
Category:
answer to question

Reply to the comments by the referee 1

  1. Most of the results and discussion are focused on the superconducting state (Fig. 1 and Fig. 2) Here the qualitative comparison between the octet model and QPI scattering is established, but this is not new information and has been discussed extensively within the literature since the early 2000’s [E.g. Mcelroy et al. (2003) -Ref 23 of the manuscript - or Pereg-Barnea et al. (2003)- Ref 37 of the manuscript]. Additionally, the “new” scattering vectors that the authors identify outside the superconducting state appear to be the expected QPI pattern from the normal state, which again is not new information [see e.g Kreisel et al. PRL 114, 217002 (2015) - not cited - or Gu et al. Nat. Comm., 10, 1603 (2019) – not cited].

Reply: We began with results at zero temperature for completeness where we have presented the results in Fig. 1 and 2 due to non-magnetic and magnetic impurities for both cases $gmap$ and $zmap$, not considered in details in earlier works. It may be noted that Monte-Carlo based approach adopted to examine the finite temperature spectral properties reduces to what is obtained within Hartree-Fock theory at low temperature as the thermal fluctuations reduces with a decline in temperature. Secondly, the thermal equilibration process at $T \sim 0$ is very slow and therefore we used Eq. 13. However, the remaining results including Fig. 3 are obtained using Eq. 1. In the pseudogap phase, there is an important difference from the normal state QPI pattern is the survival of the pattern corresponding to the scattering vectors ${\bf q_3}$ and ${\bf q_7}$ unlike ${\bf q_8}$ and ${\bf q_9}$ associated with the normal states.

We have added the new references pointed out by the referee, which were missed in the last version.

  1. The actual study of the consequence of the pseudogap on the QPI pattern appears to be missing. For example, How does the QPI pattern evolve at the Fermi level as a function of temperature? Are there clear signatures of the pseudogap in the QPI that aren't found in the normal state? Are there any changes to the real space QPI pattern as a function of temperature? From the two panels (Fig. 3(h) and 3(k)) it looks as if the pseudogap phase produces a similar QPI pattern to the normal state (Fig. 3f and 3(i)), but there is only limited information to tell if this is true or not. As this was the motivation of the study, I feel that the opportunity to discuss the new physics that this model contains was missed.

Reply: In principle, there are several differences in the QPI patterns of pseudogap when compared with those in the normal state or low-temperature superconducting states. First, the patterns are relatively sharper at the low temperature because of the absence of the spatial and thermal fluctuations in the amplitude and phase of the superconducting parameters. Secondly, while the pseudogap is accompanied with Fermi arc, the antinodal gap survives to a temperature higher than the onset temperature of $d$-wave superconductivity. Thus, an important difference from the normal state QPI pattern is the survival of the pattern corresponding to the scattering vectors ${\bf q_3}$ and ${\bf q_7}$ unlike ${\bf q_8}$ and ${\bf q_9}$ associated with the normal states.

We have added this points in the section "results and discussion '' as well as above mentioned references are also cited in the revised manuscript.

  1. I also question why the authors did not use the continuum transformation of the Green's function to study the QPI pattern? Here the authors use a Green's function that in real space would sample the density of states once per unit cell. However STM measurements are continuous in real space. The Continuum QPI (cQPI) technique has become well established in recent years, and provides a much more accurate description of the QPI seen in STM (See for example Kreisel et al. (2015) - PRL 114, 217002 - or Boker et al. (2020) - Ref 35 of this manuscript - for cQPI calculations on models of superconducting cuprates.)

Reply: Recent works have highlighted the importance of using continuum Green's function within a Wannier basis in order to take into account the electronic cloud around the lattice point with an appropriate phase associated with a particular orbital such as $d_{x^2-y^2}$ orbital in cuprates. QPI based on such a realistic description of electronic cloud provides a better description of the STM results as pointed out already in several earlier works. However, our focus, in this work, was to primarily examine the temperature dependence of QPI. The spatial distribution of the electronic cloud is going to be largely unaffected with a rise in temperature. And, the essential difference between the cases with and without the electronic clouds at low temperature has already been pointed out in earlier works (Kreisel et al. (2015) and Boker et al. (2020)), which is expected to be present even at finite temperature. It may, however, be interesting to incorporate continuum effect for the purpose of completeness in future such studies.

We have added this point in the section "Model and Method" of revised manuscript.

Modifications suggested by the referee 1

  1. The authors write that "the QPIs in the d-wave state can also be examined at low temperature by ignoring the scalar term in the Hamiltonian given in Eq. 1 and by Fourier transforming the electron creation and annihilation operator, which leads to (Eq 13)…” Here I am confused, are the calculations in the main text using the new Hamiltonian (Eq. 1) or Eq. 13. Because if they are using Eq. 13 then the QPI calculation has already been performed in multiple publications [see the four references cited above in the report for a non-exhaustive list]. If you did not use Eq. 13, then why did you write this in the methodology section? - Could you please clarify this point?

Reply: It may be noted that Monte-Carlo based approach adopted to examine the finite temperature spectral properties reduces to Hartree-Fock theory at low temperature as the thermal fluctuations reduce on lowering the temperature. Secondly, the thermal equilibration process near $T \sim 0K$ is rather slow and therefore we used Eq. 13 instead. However, the remaining results including Fig. 3 are obtained using Eq. 1.

  1. The authors refer to specific vectors (q1-q7) in the octet model throughout this text, but a definition of the vectors does not appear until Fig 4a. As a reader, I was confused as to what scattering vectors you were referring to in the QPI patterns. For example, in the caption of Fig.1 The authors write “The QPI patterns are dominated by the scattering vectors q2,q3,q6,q7” but it is not specified where q2,q3,q6,q7 belong on the g- or z-maps. Arrows highlighting the specific vectors would help the reader to understand what the authors mean.

Reply: We have modified the Figs. 1 and 2 so that the scattering vectors ${\bf q}_1, {\bf q}_2, {\bf q}_3$, ${\bf q}_5$, ${\bf q}_6,$ and ${\bf q}_7$ are now shown in addition to Fig. 4.

  1. In the results section (second paragraph page 3) “The features due to q2,q3,q6,q7 can be seen clearly when Delta w ~ 0.5 Delta. Do you mean when comparing energy layers spaced by 0.5 Delta? or do you mean when the energy is half the superconducting gap? Could you clarify what you mean?

Reply: Through the sentence, we meant that the QPI features corresponding to the scattering vectors ${\bf q}_2, {\bf q}_3, {\bf q}_6,$ and ${\bf q}_7$ can be noticed when the quasiparticle energy is near $ 0.5\Delta_0$, \textit{i. e.} $\omega \sim 0.5\Delta_0$ or quasiparticle energy is half of the superconducting gap at the antinodes.

  1. In the final paragraph of page 3 “Note the suppression of patterns because of other scattering vectors for the very reason that they don’t connect sections of CEC having sign of d-wave superconducting order parameters opposite to each other”. Doesn’t make sense, and I am unclear what the authors intend to say here. - Could you clarify?

Reply: Through the sentence, we wanted to describe the observation that the QPI patterns corresponding to the scattering vectors such as ${\bf q}_2$, ${\bf q}_3$, ${\bf q}_6$ and ${\bf q}_7$, which connect those regions of CECs with opposite sign of the superconducting order parameter, are suppressed in the presence of a magnetic impurity. The referred sentence is revised so that it can clearly convey it's message.

  1. There is some repetition in the introduction, “In this paper, we examine the QPI features of pseudogap phase at finite temperature, … …. In this paper we study the temperature dependent behavior of QPIs in a minimal model of d-wave superconductivity”.

Reply: We will remove the repeated sentence, \textit{i.e.} ``In this paper we study the temperature dependent behavior of QPIs in a minimal model of $d$-wave superconductivity'' in the revised version.

  1. Figure 4, (d),(e) and (f) are not labelled in the caption nor are they referred to in the main text. - Could you please add these descriptions to the caption and discuss the panels in the text. } \

Reply: In the revised manuscript, Fig. 4(d), (e) and (f) will be labeled as well as referred to at appropriate places in the section ``Results and discussion''. It may be noted that the discussion related to figure 4 was already included in the previous version.

Author:  Dheeraj Kumar Singh  on 2022-12-19  [id 3154]

(in reply to Report 1 on 2022-11-09)

  1. Most of the results and discussion are focused on the superconducting state (Fig. 1 and Fig. 2) Here the qualitative comparison between the octet model and QPI scattering is established, but this is not new information and has been discussed extensively within the literature since the early 2000’s [E.g. Mcelroy et al. (2003) -Ref 23 of the manuscript - or Pereg-Barnea et al. (2003)- Ref 37 of the manuscript]. Additionally, the “new” scattering vectors that the authors identify outside the superconducting state appear to be the expected QPI pattern from the normal state, which again is not new information [see e.g Kreisel et al. PRL 114, 217002 (2015) - not cited - or Gu et al. Nat. Comm., 10, 1603 (2019) – not cited].

Reply: We began with results at zero temperature for completeness where we have presented the results in Fig. 1 and 2 due to non-magnetic and magnetic impurities for both cases $gmap$ and $zmap$, not considered in details in earlier works. It may be noted that Monte-Carlo based approach adopted to examine the finite temperature spectral properties reduces to what is obtained within Hartree-Fock theory at low temperature as the thermal fluctuations reduces with a decline in temperature. Secondly, the thermal equilibration process at $T \sim 0$ is very slow and therefore we used Eq. 13. However, the remaining results including Fig. 3 are obtained using Eq. 1. In the pseudogap phase, there is an important difference from the normal state QPI pattern is the survival of the pattern corresponding to the scattering vectors ${\bf q_3}$ and ${\bf q_7}$ unlike ${\bf q_8}$ and ${\bf q_9}$ associated with the normal states.

We have added the new references pointed out by the referee, which were missed in the last version.

  1. The actual study of the consequence of the pseudogap on the QPI pattern appears to be missing. For example, How does the QPI pattern evolve at the Fermi level as a function of temperature? Are there clear signatures of the pseudogap in the QPI that aren't found in the normal state? Are there any changes to the real space QPI pattern as a function of temperature? From the two panels (Fig. 3(h) and 3(k)) it looks as if the pseudogap phase produces a similar QPI pattern to the normal state (Fig. 3f and 3(i)), but there is only limited information to tell if this is true or not. As this was the motivation of the study, I feel that the opportunity to discuss the new physics that this model contains was missed.

Reply: In principle, there are several differences in the QPI patterns of pseudogap when compared with those in the normal state or low-temperature superconducting states. First, the patterns are relatively sharper at the low temperature because of the absence of the spatial and thermal fluctuations in the amplitude and phase of the superconducting parameters. Secondly, while the pseudogap is accompanied with Fermi arc, the antinodal gap survives to a temperature higher than the onset temperature of $d$-wave superconductivity. Thus, an important difference from the normal state QPI pattern is the survival of the pattern corresponding to the scattering vectors ${\bf q_3}$ and ${\bf q_7}$ unlike ${\bf q_8}$ and ${\bf q_9}$ associated with the normal states.

We have added this points in the section ``results and discussion section'' as well as above mentioned references are also cited in the revised manuscript.

  1. I also question why the authors did not use the continuum transformation of the Green's function to study the QPI pattern? Here the authors use a Green's function that in real space would sample the density of states once per unit cell. However STM measurements are continuous in real space. The Continuum QPI (cQPI) technique has become well established in recent years, and provides a much more accurate description of the QPI seen in STM (See for example Kreisel et al. (2015) - PRL 114, 217002 - or Boker et al. (2020) - Ref 35 of this manuscript - for cQPI calculations on models of superconducting cuprates.)

Reply: Recent works have highlighted the importance of using continuum Green's function within a Wannier basis in order to take into account the electronic cloud around the lattice point with an appropriate phase associated with a particular orbital such as $d_{x^2-y^2}$ orbital in cuprates. QPI based on such a realistic description of electronic cloud provides a better description of the STM results as pointed out already in several earlier works. However, our focus, in this work, was to primarily examine the temperature dependence of QPI. The spatial distribution of the electronic cloud is going to be largely unaffected with a rise in temperature. And, the essential difference between the cases with and without the electronic clouds at low temperature has already been pointed out in earlier works (Kreisel et al. (2015) and Boker et al. (2020)), which is expected to be present even at finite temperature. It may, however, be interesting to incorporate continuum effect for the purpose of completeness in future such studies.

We have added this point in the section ``Model and Method '' of revised manuscript.

Modifications suggested by the referee

  1. The authors write that "the QPIs in the d-wave state can also be examined at low temperature by ignoring the scalar term in the Hamiltonian given in Eq. 1 and by Fourier transforming the electron creation and annihilation operator, which leads to (Eq 13)…” Here I am confused, are the calculations in the main text using the new Hamiltonian (Eq. 1) or Eq. 13. Because if they are using Eq. 13 then the QPI calculation has already been performed in multiple publications [see the four references cited above in the report for a non-exhaustive list]. If you did not use Eq. 13, then why did you write this in the methodology section? - Could you please clarify this point?

Reply: It may be noted that Monte-Carlo based approach adopted to examine the finite temperature spectral properties reduces to Hartree-Fock theory at low temperature as the thermal fluctuations reduce on lowering the temperature. Secondly, the thermal equilibration process near $T \sim 0K$ is rather slow and therefore we used Eq. 13 instead. However, the remaining results including Fig. 3 are obtained using Eq. 1.

  1. The authors refer to specific vectors (q1-q7) in the octet model throughout this text, but a definition of the vectors does not appear until Fig 4a. As a reader, I was confused as to what scattering vectors you were referring to in the QPI patterns. For example, in the caption of Fig.1 The authors write “The QPI patterns are dominated by the scattering vectors q2,q3,q6,q7” but it is not specified where q2,q3,q6,q7 belong on the g- or z-maps. Arrows highlighting the specific vectors would help the reader to understand what the authors mean.

Reply: We have modified the Figs. 1 and 2 so that the scattering vectors ${\bf q}_1, {\bf q}_2, {\bf q}_3$, ${\bf q}_5$, ${\bf q}_6,$ and ${\bf q}_7$ are now shown in addition to Fig. 4.

  1. In the results section (second paragraph page 3) “The features due to q2,q3,q6,q7 can be seen clearly when Delta w ~ 0.5 Delta. Do you mean when comparing energy layers spaced by 0.5 Delta? or do you mean when the energy is half the superconducting gap? Could you clarify what you mean?

Reply: Through the sentence, we meant that the QPI features corresponding to the scattering vectors ${\bf q}_2, {\bf q}_3, {\bf q}_6,$ and ${\bf q}_7$ can be noticed when the quasiparticle energy is near $ 0.5\Delta_0$, \textit{i. e.} $\omega \sim 0.5\Delta_0$ or quasiparticle energy is half of the superconducting gap at the antinodes.

  1. In the final paragraph of page 3 “Note the suppression of patterns because of other scattering vectors for the very reason that they don’t connect sections of CEC having sign of d-wave superconducting order parameters opposite to each other”. Doesn’t make sense, and I am unclear what the authors intend to say here. - Could you clarify?

Reply: Through the sentence, we wanted to describe the observation that the QPI patterns corresponding to the scattering vectors such as ${\bf q}_2$, ${\bf q}_3$, ${\bf q}_6$ and ${\bf q}_7$, which connect those regions of CECs with opposite sign of the superconducting order parameter, are suppressed in the presence of a magnetic impurity. The referred sentence is revised so that it can clearly convey it's message.

  1. There is some repetition in the introduction, “In this paper, we examine the QPI features of pseudogap phase at finite temperature, … …. In this paper we study the temperature dependent behavior of QPIs in a minimal model of d-wave superconductivity”.

Reply: We will remove the repeated sentence, \textit{i.e.} ``In this paper we study the temperature dependent behavior of QPIs in a minimal model of $d$-wave superconductivity'' in the revised version.

  1. Figure 4, (d),(e) and (f) are not labelled in the caption nor are they referred to in the main text. - Could you please add these descriptions to the caption and discuss the panels in the text.

Reply: In the revised manuscript, Fig. 4(d), (e) and (f) will be labeled as well as referred to at appropriate places in the section ``Results and discussion''. It may be noted that the discussion related to figure 4 was already included in the previous version.

---

## Round 1 · Referee Report · Anonymous (Referee 2) · 2022-11-25

Report

Quasiparticle interference (QPI) in superconductors is commonly studied experimentally and theoretically at low T as function of bias voltage.

The authors make an innovative proposal to investigate QPI as function of temperature to identify systematic trends of low energy quasiparticles on the transition from superconducting to pseudogap phase in d-wave materials. They show that the characteristic QPI octet pattern of the low-T superconducting state should change significantly as new characteristic scattering wave vectors come into play in the pseudogap phase. To simulate the T-dependence of the excitation energies they employ numerical methods beyond the mean field Hartree Fock approach.

Due to the innovative proposal and method the manuscript is in principle adequate for publication in SciPost.

However, possibly due to the short letter style of the manuscript there are several issues that are not well explained and need further clarification before publication:

1) The temperature dependence of all quantities derived stems exclusively form the numerical determination of gap amplitude and phases from using the Metropolis algorithm. On the other hand only the T=0 Greens functions are used in (2) and subsequent derivations rather than finite temperature Matsubara Green's function. If those were employed at some point thermal averaging with Fermi functions would appear, e.g. in the densities appearing in (11),(12). The authors should comment on this and give qualitative arguments why one can neglect this.

2) A major drawback of the presentation is that the authors only present the final result for T-dependent QPI but dont show how the excitation spectrum, i.e. gap amplitude and antinodal pseudogap change with temperature or, for that matter how the phase correlations change with T below and above Tc which they must have calculated as an input for the QPI. The temperature dependence of these quantities should be shown explicitly.

3) On several occasions the pseudogap temperature T is mentioned but it remains undefined where it is placed above Tc in the present calculation. In principle it should be visible in the quantities mentioned above in 2) as temperature where phase correlation function vanishes but the gap is still finite. The authors should clarify to which extent they can identify a T in their analysis.

4) Finally it is claimed that some qi are seen or suppressed with magnetic or normal scattering. This should be illustrated in a few examples of the QPI spectra of Figs.2,3 with present/suppressed scattering vectors shown as full or dotted lines. Otherwise it is very hard to be convinced about it.

  • validity: -
  • significance: -
  • originality: -
  • clarity: -
  • formatting: -
  • grammar: -

Author:  Dheeraj Kumar Singh  on 2022-12-19  [id 3153]

(in reply to Report 3 on 2022-11-25)

  1. The temperature dependence of all quantities derived stems exclusively form the numerical determination of gap amplitude and phases from using the Metropolis algorithm. On the other hand only the T=0 Greens functions are used in (2) and subsequent derivations rather than finite temperature Matsubara Green's function. If those were employed at some point thermal averaging with Fermi functions would appear, e.g. in the densities appearing in (11),(12). The authors should comment on this and give qualitative arguments why one can neglect this

Reply: The purpose of presenting the Green's function defined by Eq. 2 was to illustrate how the Green's function in the momentum space is obtained if the spatially fluctuating complex fields are given. It is obtained using $\langle u_{\alpha l} |$ and $\langle v_{\alpha l} |$, which form an eigenvector of the Bogoliubov-de Gennes Hamiltonian corresponding to Eq. 1 for an eigenvalue $E_{\alpha l}$. The subscript $\alpha$ indicates a particular lattice site while $l$ identifies a particular lattice in the superlattice structure.

At this point, it may be noted, however, that for evaluating the Green's function or it's modification due to an impurity atom at a given temperature makes use of the thermally equilibrated configuration of the complex fields \Delta^{\delta}_{ i} in the Bogoliubov-de Gennes Hamiltonian corresponding to Eq. 1. The thermally equilibrated classical amplitude and phase fields are generated in accordance with the probability distribution $P{\Delta_i, \phi^x_i, \phi^y_i } \propto Tr_{dd^{\dagger}}e^{-\beta H_{eff}} $. The latter involves Fermionic tracing which cannot be evaluated exactly. Thus, we use Metropolis algorithm through which all the types of thermal fluctuations are incorporated. Then, the Eq. 3 is used to calculate the finite-temperature single-particle spectral function. The calculation is repeated several times at a given temperature and then the average is evaluated, which gives thermally averaged spectral function. Thus, the role of temperature is already incorporated into this scheme.

  1. A major drawback of the presentation is that the authors only present the final result for T-dependent QPI but dont show how the excitation spectrum, i.e. gap amplitude and antinodal pseudogap change with temperature or, for that matter how the phase correlations change with T below and above Tc which they must have calculated as an input for the QPI. The temperature dependence of these quantities should be shown explicitly.

Reply: In a separate work, the behavior of the single-particle spectral function of the model given by Eq. 1 has already been examined in details. For the nearest-neighbor attractive parameter $V \sim 1$ as also in the current work, the long-range phase correlation was found to develop near $T \sim T_c$, whereas the short-range phase correlation continued to exist up to a even higher temperature. The antinodal gap persisted beyond the superconducting transition temperature $T_c$. It was shown that for $T < T_c$, despite the presence of thermal phase fluctuations, the superconductor has only nodal Fermi points while all non-nodal points on the normal state Fermi surface show a two-peak spectral function with a dip at $\omega = 0$. For $T > T_c$, the Fermi points develop into arcs, characterized by a single quasiparticle peak, and the arcs connect up to recover the normal state Fermi surface at a temperature $T*> T_c$. In the current work, we present only the constant energy surfaces at different temperatures for a given quasiparticle energy. The shape of constant energy surfaces also indicate the nature of gap structure along the normal-state Fermi surface. For $V\sim1$, as in the current work, $T^* \sim 1.5T_c$.

We have incorporated a discussion about various temperature-dependent spectral features with the model given by Eq. 1 in the manuscript in ``Model and Method'' section.

  1. On several occasions the pseudogap temperature T is mentioned but it remains undefined where it is placed above Tc in the present calculation. In principle it should be visible in the quantities mentioned above in 2) as temperature where phase correlation function vanishes but the gap is still finite. The authors should clarify to which extent they can identify a T in their analysis.

Reply: As mentioned in the reply to the previous question, the onset temperature of the pseudogap phase is $T^* \sim 1.5T_c$ for the parameter $V \sim 1$ chosen in this work. We have clarified it in the revised manuscript.

  1. Finally it is claimed that some qi are seen or suppressed with magnetic or normal scattering. This should be illustrated in a few examples of the QPI spectra of Figs.2,3 with present/suppressed scattering vectors shown as full or dotted lines. Otherwise it is very hard to be convinced about it.

Reply: In the Figs. 2 and 3 of the revised manuscript, we have used arrows to highlight such scattering vectors.

---

## Round 1 · Referee Report · Anonymous (Referee 3) · 2022-12-1

Report

The main problem I see is that the authors did not elaborate on the subject siginificantly. I assume that the main temperature dependence should be near the Fermi level, where the contribution of the thermally excited quasiparticles is important. At the same time the authors mostly show QPI maps for various bias energies for a given temperature and it is not clear what do they actually want to study. Thus, it is hard to judge whether and where new physics arises. For example, I expect some crossover omega \sim T , which determines on the importance of the temperature-based effects in the qpi. So in my opinion, only looking for biases omega<T makes sense if one wants to study the temperature dependent effects.

I also think there is further problem related to the fact that the tunneling is taking place via apical oxygen, which requires to use continuous Green's function to model QPI in cuprates, so I wonder whether this will affect the present conclusions.
  • validity: ok
  • significance: ok
  • originality: ok
  • clarity: low
  • formatting: good
  • grammar: good

Author:  Dheeraj Kumar Singh  on 2022-12-19  [id 3152]

(in reply to Report 2 on 2022-12-01)
Category:
answer to question

  1. The main problem I see is that the authors did not elaborate on the subject siginificantly. I assume that the main temperature dependence should be near the Fermi level, where the contribution of the thermally excited quasiparticles is important. At the same time the authors mostly show QPI maps for various bias energies for a given temperature and it is not clear what do they actually want to study. Thus, it is hard to judge whether and where new physics arises. For example, I expect some crossover omega $\sim T$ , which determines on the importance of the temperature-based effects in the qpi. So in my opinion, only looking for biases omega<T makes sense if one wants to study the temperature dependent effects.

    Reply: It is indeed true that the temperature-based effect on the QPI are expected to be prominent when the quasiparticle energy $\omega \sim T$. A significant effect is that the sharp QPI features will get thermally broadened. In addition, the momentum-dependent gap structure along the normal state Fermi surface may also exhibit a rather complex temperature dependence as shown by the ARPES measurements. In particular, a remarkable modification in the Fermiology takes place, where the Fermi points turns into Fermi arc accompanied with a persisting antinodal gap beyond the superconducting transition temperature. Several of these features were reproduced in an earlier work with an approach adopted in the current work. We were largely interested in obtaining those dominant features and associated scattering vectors. We will provide additional discussion in order to make these points more readily available to the readers.

  2. I also think there is further problem related to the fact that the tunneling is taking place via apical oxygen, which requires to use continuous Green's function to model QPI in cuprates, so I wonder whether this will affect the present conclusions.

Reply: This is indeed true that the QPI based on such a realistic description of electronic cloud implemented through continous Green's function provides a better description of the STM results as pointed out in several recent works. However, our focus, in the current work, was to primarily examine the temperature dependence of QPI. We believe the continuum Green's function consideration is expected to have not any major role in the temperature dependence of QPI. This is because the spatial distribution of the electronic cloud is going to be largely unaffected to the first order of approximation. And, the essential difference between the cases with and without the electronic clouds at low temperature has already been pointed out in earlier works (Kreisel et al. (2015) and Boker et al. (2020)). It may, however, be interesting to incorporate continuum effect for the purpose of completeness in future such studies.

 We have added a discussion based on this point in the revised manuscript.

---

## Round 2 · Referee Report · Anonymous · 2022-12-22

Report

The authors have adequately responded to my
previous objections and suggestions and have
made according changes in the manuscript.
I think in its present form it fulfils the general
requirements for publication in SciPost

---

## Round 2 · Referee Report · Anonymous · 2023-1-5

Report

The authors have sufficiently addressed my concerns with the first submission, as well as the points raised by the other referees. I believe this manuscript is now suitable for publication in SciPost Physics Core.

---

## Round 2 · Author Response

(C) Response to the referee comments on 2210.10521v1

We would like to thank the referees for the valuable comments as well as suggestions. We have made several changes to the manuscript as pointed out by the referee. Point-wise reply to the comments are as below.

Reply to the comments by the referee 1

  1. Most of the results and discussion are focused on the superconducting state (Fig. 1 and Fig. 2) Here the qualitative comparison between the octet model and QPI scattering is established, but this is not new information and has been discussed extensively within the literature since the early 2000’s [E.g. Mcelroy et al. (2003) -Ref 23 of the manuscript - or Pereg-Barnea et al. (2003)- Ref 37 of the manuscript]. Additionally, the “new” scattering vectors that the authors identify outside the superconducting state appear to be the expected QPI pattern from the normal state, which again is not new information [see e.g Kreisel et al. PRL 114, 217002 (2015) - not cited - or Gu et al. Nat. Comm., 10, 1603 (2019) – not cited].

Reply: We began with results at zero temperature for completeness where we have presented the results in Fig. 1 and 2 due to non-magnetic and magnetic impurities for both cases $gmap$ and $zmap$, not considered in details in earlier works. It may be noted that Monte-Carlo based approach adopted to examine the finite temperature spectral properties reduces to what is obtained within Hartree-Fock theory at low temperature as the thermal fluctuations reduces with a decline in temperature. Secondly, the thermal equilibration process at $T \sim 0$ is very slow and therefore we used Eq. 13. However, the remaining results including Fig. 3 are obtained using Eq. 1. In the pseudogap phase, there is an important difference from the normal state QPI pattern is the survival of the pattern corresponding to the scattering vectors ${\bf q_3}$ and ${\bf q_7}$ unlike ${\bf q_8}$ and ${\bf q_9}$ associated with the normal states.

We have added the new references pointed out by the referee, which were missed in the last version.

  1. The actual study of the consequence of the pseudogap on the QPI pattern appears to be missing. For example, How does the QPI pattern evolve at the Fermi level as a function of temperature? Are there clear signatures of the pseudogap in the QPI that aren't found in the normal state? Are there any changes to the real space QPI pattern as a function of temperature? From the two panels (Fig. 3(h) and 3(k)) it looks as if the pseudogap phase produces a similar QPI pattern to the normal state (Fig. 3f and 3(i)), but there is only limited information to tell if this is true or not. As this was the motivation of the study, I feel that the opportunity to discuss the new physics that this model contains was missed.

Reply: In principle, there are several differences in the QPI patterns of pseudogap when compared with those in the normal state or low-temperature superconducting states. First, the patterns are relatively sharper at the low temperature because of the absence of the spatial and thermal fluctuations in the amplitude and phase of the superconducting parameters. Secondly, while the pseudogap is accompanied with Fermi arc, the antinodal gap survives to a temperature higher than the onset temperature of $d$-wave superconductivity. Thus, an important difference from the normal state QPI pattern is the survival of the pattern corresponding to the scattering vectors ${\bf q_3}$ and ${\bf q_7}$ unlike ${\bf q_8}$ and ${\bf q_9}$ associated with the normal states.

We have added this points in the section "results and discussion section" as well as above mentioned references are also cited in the revised manuscript.

  1. I also question why the authors did not use the continuum transformation of the Green's function to study the QPI pattern? Here the authors use a Green's function that in real space would sample the density of states once per unit cell. However STM measurements are continuous in real space. The Continuum QPI (cQPI) technique has become well established in recent years, and provides a much more accurate description of the QPI seen in STM (See for example Kreisel et al. (2015) - PRL 114, 217002 - or Boker et al. (2020) - Ref 35 of this manuscript - for cQPI calculations on models of superconducting cuprates.)

Reply: Recent works have highlighted the importance of using continuum Green's function within a Wannier basis in order to take into account the electronic cloud around the lattice point with an appropriate phase associated with a particular orbital such as $d_{x^2-y^2}$ orbital in cuprates. QPI based on such a realistic description of electronic cloud provides a better description of the STM results as pointed out already in several earlier works. However, our focus, in this work, was to primarily examine the temperature dependence of QPI. The spatial distribution of the electronic cloud is going to be largely unaffected with a rise in temperature. And, the essential difference between the cases with and without the electronic clouds at low temperature has already been pointed out in earlier works (Kreisel et al. (2015) and Boker et al. (2020)), which is expected to be present even at finite temperature. It may, however, be interesting to incorporate continuum effect for the purpose of completeness in future such studies.

We have added this point in the section "Model and Method" of revised manuscript.

Modifications suggested by the referee

  1. The authors write that "the QPIs in the d-wave state can also be examined at low temperature by ignoring the scalar term in the Hamiltonian given in Eq. 1 and by Fourier transforming the electron creation and annihilation operator, which leads to (Eq 13)…” Here I am confused, are the calculations in the main text using the new Hamiltonian (Eq. 1) or Eq. 13. Because if they are using Eq. 13 then the QPI calculation has already been performed in multiple publications [see the four references cited above in the report for a non-exhaustive list]. If you did not use Eq. 13, then why did you write this in the methodology section? - Could you please clarify this point?

Reply: It may be noted that Monte-Carlo based approach adopted to examine the finite temperature spectral properties reduces to Hartree-Fock theory at low temperature as the thermal fluctuations reduce on lowering the temperature. Secondly, the thermal equilibration process near $T \sim 0K$ is rather slow and therefore we used Eq. 13 instead. However, the remaining results including Fig. 3 are obtained using Eq. 1.

  1. The authors refer to specific vectors (q1-q7) in the octet model throughout this text, but a definition of the vectors does not appear until Fig 4a. As a reader, I was confused as to what scattering vectors you were referring to in the QPI patterns. For example, in the caption of Fig.1 The authors write “The QPI patterns are dominated by the scattering vectors q2,q3,q6,q7” but it is not specified where q2,q3,q6,q7 belong on the g- or z-maps. Arrows highlighting the specific vectors would help the reader to understand what the authors mean.

Reply: We have modified the Figs. 1 and 2 so that the scattering vectors ${\bf q}_1, {\bf q}_2, {\bf q}_3$, ${\bf q}_5$, ${\bf q}_6,$ and ${\bf q}_7$ are now shown in addition to Fig. 4.

  1. In the results section (second paragraph page 3) “The features due to q2,q3,q6,q7 can be seen clearly when Delta w ~ 0.5 Delta. Do you mean when comparing energy layers spaced by 0.5 Delta? or do you mean when the energy is half the superconducting gap? Could you clarify what you mean?

Reply: Through the sentence, we meant that the QPI features corresponding to the scattering vectors ${\bf q}_2, {\bf q}_3, {\bf q}_6,$ and ${\bf q}_7$ can be noticed when the quasiparticle energy is near $ 0.5\Delta_0$, \textit{i. e.} $\omega \sim 0.5\Delta_0$ or quasiparticle energy is half of the superconducting gap at the antinodes.

  1. In the final paragraph of page 3 “Note the suppression of patterns because of other scattering vectors for the very reason that they don’t connect sections of CEC having sign of d-wave superconducting order parameters opposite to each other”. Doesn’t make sense, and I am unclear what the authors intend to say here. - Could you clarify?

Reply: Through the sentence, we wanted to describe the observation that the QPI patterns corresponding to the scattering vectors such as ${\bf q}_2$, ${\bf q}_3$, ${\bf q}_6$ and ${\bf q}_7$, which connect those regions of CECs with opposite sign of the superconducting order parameter, are suppressed in the presence of a magnetic impurity. The referred sentence is revised so that it can clearly convey it's message.

  1. There is some repetition in the introduction, “In this paper, we examine the QPI features of pseudogap phase at finite temperature, … …. In this paper we study the temperature dependent behavior of QPIs in a minimal model of d-wave superconductivity”.

Reply: We have removed the repeated sentence, \textit{i.e.} "In this paper we study the temperature dependent behavior of QPIs in a minimal model of $d$-wave superconductivity" in the revised version.

  1. Figure 4, (d),(e) and (f) are not labelled in the caption nor are they referred to in the main text. - Could you please add these descriptions to the caption and discuss the panels in the text.

Reply: In the revised manuscript, Fig. 4(d), (e) and (f) are labeled as well as referred to at appropriate places in the section "Results and discussion". It may be noted that the discussion related to figure 4 was already included in the previous version.

Reply to the comments by the referee 2

  1. The main problem I see is that the authors did not elaborate on the subject siginificantly. I assume that the main temperature dependence should be near the Fermi level, where the contribution of the thermally excited quasiparticles is important. At the same time the authors mostly show QPI maps for various bias energies for a given temperature and it is not clear what do they actually want to study. Thus, it is hard to judge whether and where new physics arises. For example, I expect some crossover omega $\sim T$ , which determines on the importance of the temperature-based effects in the qpi. So in my opinion, only looking for biases omega<T makes sense if one wants to study the temperature dependent effects.

Reply: It is indeed true that the temperature-based effect on the QPI are expected to be prominent when the quasiparticle energy $\omega \sim T$. A significant effect is that the sharp QPI features will get thermally broadened. In addition, the momentum-dependent gap structure along the normal state Fermi surface may also exhibit a rather complex temperature dependence as shown by the ARPES measurements. In particular, a remarkable modification in the Fermiology takes place, where the Fermi points turns into Fermi arc accompaneid with a persisting antinodal gap beyond the superconducting transition temperature. Several of these features were reproduced in an earlier work with an approach adopted in the current work. We were largely interested in obtaining those dominant features and associated scattering vectors. We now provide additional discussion in order to make these points more readily available to the readers.

  1. I also think there is further problem related to the fact that the tunneling is taking place via apical oxygen, which requires to use continuous Green's function to model QPI in cuprates, so I wonder whether this will affect the present conclusions.

Reply: This is indeed true that the QPI based on such a realistic description of electronic cloud implemented through continous Green's function provides a better description of the STM results as pointed out in several recent works. However, our focus, in the current work, was to primarily examine the temperature dependence of QPI. We believe the continuum Green's function consideration is expected to have not any major role in the temperature dependence of QPI. This is because the spatial distribution of the electronic cloud is going to be largely unaffected to the first order of approximation. And, the essential difference between the cases with and without the electronic clouds at low temperature has already been pointed out in earlier works (Kreisel et al. (2015) and Boker et al. (2020)). It may, however, be interesting to incorporate continuum effect for the purpose of completeness in future such studies.

 We have added a discussion based on this point in the revised manuscript.

Reply to the comments by the referee 3

  1. The temperature dependence of all quantities derived stems exclusively form the numerical determination of gap amplitude and phases from using the Metropolis algorithm. On the other hand only the T=0 Greens functions are used in (2) and subsequent derivations rather than finite temperature Matsubara Green's function. If those were employed at some point thermal averaging with Fermi functions would appear, e.g. in the densities appearing in (11),(12). The authors should comment on this and give qualitative arguments why one can neglect this

Reply:The purpose of presenting the Green's function defined by Eq. 2 was to illustrate how the Green's function in the momentum space is obtained if the spatially fluctuating complex fields are given. It is obtained using $\langle u_{\alpha l} |$ and $\langle v_{\alpha l} |$, which form an eigenvector of the Bogoliubov-de Gennes Hamiltonian corresponding to Eq. 1 for an eigenvalue $E_{\alpha l}$. The subscript $\alpha$ indicates a particular lattice site while $l$ identifies a particular lattice in the superlattice structure.

At this point, it may be noted, however, that for evaluating the Green's function or it's modification due to an impurity atom at a given temperature makes use of the thermally equilibrated configuration of the complex fields $\Delta_{\mathbf{i}}^{\delta}$ in the Bogoliubov-de Gennes Hamiltonian corresponding to Eq. 1. The thermally equilibrated classical amplitude and phase fields are generated in accordance with the probability distribution $P{\Delta_i, \phi^x_i, \phi^y_i } \propto Tr_{dd^{\dagger}}e^{-\beta H_{eff}} $. The latter involves Fermionic tracing which cannot be evaluated exactly. Thus, we use Metropolis algorithm through which all the types of thermal fluctuations are incorporated. Then, the Eq. 3 is used to calculate the finite-temperature single-particle spectral function. The calculation is repeated several times at a given temperature and then the average is evaluated, which gives thermally averaged spectral function. Thus, the role of temperature is already incorporated into this scheme.

  1. A major drawback of the presentation is that the authors only present the final result for T-dependent QPI but dont show how the excitation spectrum, i.e. gap amplitude and antinodal pseudogap change with temperature or, for that matter how the phase correlations change with T below and above Tc which they must have calculated as an input for the QPI. The temperature dependence of these quantities should be shown explicitly.

Reply: In a separate work, the behavior of the single-particle spectral function of the model given by Eq. 1 has already been examined in details. For the nearest-neighbor attractive parameter $V \sim 1$ as also in the current work, the long-range phase correlation was found to develop near $T \sim T_c$, whereas the short-range phase correlation continued to exist up to a even higher temperature. The antinodal gap persisted beyond the superconducting transition temperature $T_c$. It was shown that for $T < T_c$, despite the presence of thermal phase fluctuations, the superconductor has only nodal Fermi points while all non-nodal points on the normal state Fermi surface show a two-peak spectral function with a dip at $\omega = 0$. For $T > T_c$, the Fermi points develop into arcs, characterized by a single quasiparticle peak, and the arcs connect up to recover the normal state Fermi surface at a temperature $T^* > T_c$. In the current work, we present only the constant energy surfaces at different temperatures for a given quasiparticle energy. The shape of constant energy surfaces also indicate the nature of gap structure along the normal-state Fermi surface. For $V\sim1$, as in the current work, $T^* \sim 1.5T_c$.

We have incorporated a discussion about various temperature-dependent spectral features with the model given by Eq. 1 in the manuscript in "Model and Method" section.

  1. On several occasions the pseudogap temperature T is mentioned but it remains undefined where it is placed above Tc in the present calculation. In principle it should be visible in the quantities mentioned above in 2) as temperature where phase correlation function vanishes but the gap is still finite. The authors should clarify to which extent they can identify a T in their analysis.

Reply: As mentioned in the reply to the previous question, the onset temperature of the pseudogap phase is $T^* \sim 1.5T_c$ for the parameter $V \sim 1$ chosen in this work. We have clarified it in the revised manuscript.

  1. Finally it is claimed that some qi are seen or suppressed with magnetic or normal scattering. This should be illustrated in a few examples of the QPI spectra of Figs.2,3 with present/suppressed scattering vectors shown as full or dotted lines. Otherwise it is very hard to be convinced about it.

Reply: In the Figs. 2 and 3 of the revised manuscript, we have used arrows to highlight such scattering vectors.

---

## Round 2 · List of Changes

(B) The major changes to the paper are:

(i) In response the comments made by the referee 1 and 3, arrows are now used to show the scattering vectors corresponding to different QPI patterns in the Fig. 2 and 3.

(ii) The referees indicated that results were not discussed elaborately. In particular, it was not clear how the results obtained here can be used to differentiate the pseudogap phase from any other phase. In the revised manuscript, we have added points in the section ``results and discussion'' to make the discussion more readily available to the reader.

(iii) Referees 1 and 3 has raised question on not using continuum Green's function. We have provided a brief comment on it in the form of a paragraph in the section ``results and discussion''.

(iv) Referee 3 pointed out the absence of details of single-particle spectral function indicating the momentum-dependent gap structure. We have already presented the details of gap structure of the single-particle spectral elsewhere. Therefore, a pragraph describing major features of the excitation spectrum is provided in the revised manuscript.

(v) Several new references have been added to the manuscript, in
light of the new discussions and comments made by the referees.

---

## Editorial Decision

published